# Association between Physical Activity Levels and Brain Volumes in Adults Visiting Radio-Imaging Center of Tertiary Care Hospital

**DOI:** 10.3390/ijerph192417079

**Published:** 2022-12-19

**Authors:** Deepika Raja, Sneha Ravichandran, Baskaran Chandrasekaran, Rajagopal Kadavigere, M. G. Ramesh Babu, Meshari Almeshari, Amjad R. Alyahyawi, Yasser Alzamil, Ahmad Abanomy, Suresh Sukumar

**Affiliations:** 1Department of Medical Imaging Technology, MCHP, MAHE, Manipal 576104, India; 2Department of Exercise and Sports Sciences, MCHP, MAHE, Manipal 576104, India; 3Department of Radiodiagnosis and Imaging, KMC, MAHE, Manipal 576104, India; 4Department of Basic Medical Sciences, MAHE, Manipal 576104, India; 5Department of Diagnostic Radiology, College of Applied Medical Science, University of Hail, Ha’il 81442, Saudi Arabia; 6Department of Radiological Sciences, Department of Diagnostic Radiology, College of Applied Medical Science, King Saud University, P.O. Box 10219, Riyadh 11451, Saudi Arabia

**Keywords:** sedentary behaviour, prolonged sitting, Brain Size, MRI, AAL3, brain volume

## Abstract

Background and aim: There is evidence to support the favorable impact of physical activity (PA) on brain volume. However, the empirical evidence exploring the relationship between physical and sedentary behavior remains mixed. We aimed to explore the relationship between PA and sedentary behavior and brain volume. Methods: The study sample (n = 150, mean age = 39.7 years) included patients interviewed with the International Physical Activity Questionnaire (IPAQ) who underwent an MRI brain scan. From the images obtained, we measured total intracranial, gray matter, and white matter volume along with the hippocampus, amygdala, parahippocampal gyrus, and posterior cingulate cortex (PCC). Multivariable linear regression analysis was done. Results and discussion: Left hippocampus and overall PA were positively and significantly associated (β = 0.71, *p* = 0.021) whereas time spent on vigorous physical activity showed a negative association (β = −0.328, *p* = 0.049) with left hippocampal volume. Conclusion: We found a positive association between total PA and the left hippocampus, whereas vigorous PA showed a negative association with the left hippocampus.

## 1. Introduction

Any activity that consists of musculoskeletal activity that results in energy expenditure is termed as physically active [1], whereas sedentary behavior is activity that results in a low energy expenditure of less than 1.5 metabolic equivalents. Considerable evidence shows that physical inactivity (failing to meet the global recommendations of 150 minutes of moderate-vigorous PA in a week) and sedentary behavior is linked with chronic disease risk including cardiometabolic and mental health risk [2]. Though physical inactivity and sedentary behaviors are identified as potential health risks, only a fraction of the global population meets the global recommendations for PA [3]. The potential reason could be the mixed evidence on the association between sedentary behavior and insufficient activity with chronic disease risk, especially psychological and brain health [4,5,6,7,8].

Brain structure results from an optimized neural interaction between environmental and genetic factors and is related to cognitive functions and academic achievement in children, work productivity in adults, and quality of life in the elderly [9,10,11]. Binge television watching and high internet use are associated with reduced regional gray matter (cingulate, hypothalamus, insula, prefrontal, temporal and occipital areas) [11]. High sedentary time and low PA are potential risk factors for brain volume loss and cognitive decline in the elderly. Anecdotal evidence claims the favorable effects of PA and sedentary behavior interventions on cognitive functions [12,13]. However, the relationship between PA and sedentary behavior on brain volumes and cognitive decline remains unconvincing [4,14]. The relationship between various intensities of PA and sedentary time with brain volumes is yet to be explored. In this cross-sectional study, we explored the association between PA of varying volumes and sedentary time with brain volume in adults and the elderly visiting an outpatient radio-imaging department of a multidisciplinary rural hospital. 

## 2. Materials and Methods

This cross-sectional study is reported according to the Strengthening the Reporting of Observational Studies in Epidemiology (STROBE) guidelines [15]. The study was started after formal approval from the Institutional Research Committee, Manipal College of Health Professions, Institutional Ethics Committee, KH (IEC: 385/2021) and prospectively registered in the Clinical Trial Registry of India (CTRI/2021/09/036160). The study also conformed to the ethical foundations laid by the Helsinki Declaration.

### 2.1. Participants

The research was carried out between September 2021 and March 2022 at the Department of Radiodiagnosis and Imaging of a multidisciplinary tertiary care hospital. All the participants recruited for this study were patients scheduled for magnetic resonance imaging (MRI) brain scans, between 18 and 60 years old, and were willing to participate in the study. Before the start of the research, informed consent was acquired from all participants, and a thorough description of all study protocols, including MRI safety precautions, was given to them. Participants with a history of dementia, major psychiatric or neurologic disorders, drug abuse, head trauma or systemic disease negatively impacting cognitive abilities, uncontrolled high blood pressure, or cardiovascular disease were excluded from the study.

### 2.2. Procedure

The patients who were scheduled for brain MRI scans in the radiology department of a multidisciplinary teaching hospital were approached for participation in the study. The volunteering patient’s demographic characteristics (age, gender, education, marital status), cardiovascular risk factors (smoking, alcohol, and PA), and medication use were obtained.

### 2.3. Physical Activity Assessment

Seven days of self-reported PA levels were assessed using the International Physical Activity Questionnaire (IPAQ-SF) [16]. The questionnaire has four signaling questions eliciting responses regarding the duration and frequency of vigorous, moderate, and light-intensity activities with the duration of daily sitting time in hours. The PA volume from each dimension (vigorous, moderate intensity, and walking) and frequency spent on them. Metabolic Equivalent minutes per week (MET-min/week) were calculated using Ainsworth calculations and scored for intensity as walking  =  3.3 METs, moderate PA  =  4.0 METs, and vigorous PA  =  8.0 METs (Ainsworth et al. 2011).

### 2.4. Image Acquisition

Three-dimensional sets of whole-brain data for all participants were acquired on a 1.5 T General Electric Signa scanner using a standard eight-channel head coil. We acquired a T1-weighted, fast spoiled gradient echo (FSPGR) BRAVO sequence using the standard department protocols. The extrinsic image parameters used were repetition time (TR) = 9 ms, echo time (TE) = 4 ms, and flip angle of 13°, consisting of axial partitions with a slice thickness of 1.2 mm. This parameter provides high-resolution images in 120 ± 4 slices covering the whole brain.

### 2.5. Image Pre-Processing

The pre-processing steps were carried out with the help of CAT12 (computational anatomy toolbox—version 12.8) (RRID: SCR_019184 accessed on 1 March 2022 installed in the SPM12 (statistical parametric mapping—version 12 (Version7771)) software (RRID:SCR_007037) accessed on 15 April 2022 (ftp:/ftp.fil.ion.ucl.ac.uk/spm.) MATLAB (version R2019a) (RRID:SCR_001622) was used as a common platform for these programs. MRI Convert version 2.1.0 accessed on 1 March 2022 was used to convert all DICOM images to NIfTi format. During the pre-processing steps, all the images were normalized and segmented. Furthermore, all pre-processed images were subjugated to a sample homogeneity test and a careful evaluation for image quality. Images with a weighted overall image quality of more than 70% were chosen for further examination. Furthermore, using the estimate option in the software, values for volumes of GM, WM, cerebrospinal fluid (CSF), and total intracranial volume (TIV) were calculated from each image using the estimate option in the CAT12 software. GMVs of 170 brain areas based on the automated anatomical labeling (AAL-3) atlas were acquired for each participant during pre-processing steps using the CAT12 tool in expert mode. The prime page has been shown in Figure 1. 

The baseline characteristics were summarized using descriptive analysis. As the data was normal, as visualized using Q–Q plots and the Shapiro–Wilk test, we applied linear regression analysis using the enter method. The strength of association was sought between the PA volume (nominal: light, moderate, and high intensity; continuous: METmin/week) with the estimated brain volumes (continuous: hippocampal). All statistical analyses were carried out using SPSS 16.0 (SPSS Inc., Chicago, IL, USA).

## 3. Results

Of 384 participants contacted, 247 patients volunteered to participate in the study, and 150 participants’ data were included for reporting after the analysis represented in Figure 2.

### 3.1. Baseline Characteristics

The data were analyzed from 150 patients ranging in age from 18 to 60 years old, with an average age of 39.7 years. Out of 150 patients, 86 were male, and 64 were female. Lifestyle factors and cardiovascular risk factors were enquired about during the baseline assessment. Out of 150 patients assessed, 12 had hypertension, 6 had type-II diabetes mellitus, 4 had a smoking habit, and 9 had occasional alcohol consumption. Patients with hypertension and type-II diabetes mellitus were under medication. The data are tabulated in Table 1.

### 3.2. Physical Activity among the Participants

In terms of PA, the participants were categorized as low *n* = 59 (39.33%), moderate *n* = 78 (52%), and high *n* = 13 (8.67%). The average METs value in the sample was 924.46 MET-min/week. The average METs score for each category was found to be: low = 239.74 min/week, moderate = 1103.9 min/week, and high = 3162.9 min/week.

### 3.3. Association between Brain Volumes, Physical Activity, and Sedentary Behavior

Left hippocampus volume and PA (METs) were positively and significantly associated with β = 0.71, *p* = 0.021. Sedentary behavior was negatively and non-significantly associated with left hippocampal volume with β = −0.053, *p* = 0.586. However, right-side hippocampal thickness did not show any significant association with physical activity (METs) and sitting time with β = 0.564, *p* = 0.079 and β = −0.002, *p* = 0.988, respectively. On comparing the individual effect of low, moderate, and vigorous PA, the left hippocampus showed a negative association with vigorous PA with β = −0.328, *p* = 0.049. But the right hippocampus did not show any such relation with time spent in different intensities of PA. The data are tabulated in Table 2.

## 4. Discussion

The main aim of this study was to demonstrate the relationship between PA, sedentary behavior, and cerebral grey matter volume. The relationship between PA, sedentary behavior, total intracranial volume, grey matter volume, and white matter volume was investigated. Additionally, region-specific analyses were performed on the hippocampus, parahippocampus, PCC, and amygdala. According to the World Health Organization, people aged 18–64 years should participate in at least 150–300 min of moderate-intensity aerobic PA or 75–150 min of vigorous-intensity aerobic PA per week, or an equivalent combination [17]. In our present study, we observed that about 39.33% of the participants were in the low, 52% in the moderate, and 8.67% in the high PA category. Most of the participants did not meet the WHO guidelines with regard to the time spent in moderate or vigorous PA.

In recent years, researchers and policymakers have focused their research on fostering brain health or healthy brain aging, particularly among the elderly. PA has been suggested as a simple lifestyle method to counteract age-related brain atrophy, supported by a growing body of evidence. Various observational epidemiological studies have investigated the potential relationship between PA and cognitive decline, dementia, and Alzheimer’s disease, with varying results depending on research methodology (e.g., follow-up time), disparities in cohort demographics, adjustment for confounders, and PA assessment [18,19,20,21,22].

Many studies have found a link between brain volume and several types or intensities of PA. According to Varma et al., in a sample of non-demented, preponderantly physically inactive older people, increased walking on a daily basis was linked with a higher level of hippocampus volumes among the older female population [23]. They also noted that, compared to the control brain region, thalamus volume, this link was limited to hippocampus volume and remained significant for low-intensity walking activity, irrespective of moderate- to vigorous-intensity activity or self-reported exercise.

The hippocampus is known to have a role in learning and memory and spatial navigation, emotional behavior, and hypothalamic function regulation. The current study’s findings show a significant positive relationship between the left hippocampus and PA. We found that the left hippocampus was positively associated with total PA. This observation is consistent with previous research. A one-year exercise intervention study done by Erickson et al. showed that aerobic exercise was found to be beneficial in increasing the size of the hippocampus. The study also suggests that the effect of exercise was rather selective, affecting only the anterior hippocampus and neither the thalamus nor the caudate nucleus, based on the several regions they had examined. This suggests that exercise does not have the same effect on all brain regions [24]. Similarly, Ruotsalainen I et al. discovered that aerobic fitness, but not PA, is related to grey matter volume [25].

Additionally, we found a significant negative association between vigorous PA and the left hippocampus. We have considered the time spent in low, moderate, and vigorous PA in the current study. Vigorous-intensity PA has not been investigated separately from moderate PA for its association with brain volume in our literature evaluation. Previous research that looked at the influence of exercise on the hippocampus volume found either positive or null results. The results from our study seem to be contradictory to the existing evidence. However, a randomized control trial conducted by Pani J et al. to show the effect of different intensity exercises on brain volume demonstrated a similar result. After a study period of five years, they found that the High-Intensity Interval Training group exhibited considerably more hippocampus atrophy in the CA1 and hippocampal body while it was still within normal limits [26].

The PCC is one of the metabolically active regions and is highly connected with other brain regions. An article published by Leech et al. talks about the significance of the posterior cingulate cortexin cognition and disease. Reduced metabolism in the PCC is a symptom of Alzheimer’s disease that typically appears before a clinical diagnosis is made [27]. A study by Voss M et al. showed that aerobic fitness is positively associated with functional connectivity of the PCC and other brain regions [28]. We could not find any significant association between the PCC and PA in the present study.

Many cognitive processes have been linked to the parahippocampal cortex (PHC), including visuospatial processing and episodic memory. Siddarth P et al. conducted a pilot study to look for the effect of PA on hippocampal sub-regions in older adults having memory complaints. Their results show that the high PA group had a greater thickness of the PHC [5]. However, in the present study, we could not find an association between parahippocampal volume and PA.

It is well-known that memory function is reliant on the hippocampus and amygdala, and atrophy in these areas is an early indication of Alzheimer’s disease. A study by Moored et al. identified that partaking in a wider range of activities as a youth was linked to greater amygdala dimensions in late life [29]. In the current study, we were unable to find a significant relationship between PA and the amygdala.

Our study findings failed to observe any significant association between sedentary behavior and brain volume, although there was a negative association between all brain ROIs except for the right PHC. The impact of sedentary behavior on a person’s health is well known. Sedentary behavior is linked to an elevated risk of diabetes, hypertension, and obesity, and a reduction in vascular supply [30]. Sedentary behavior was attributed to less thickness in the MTL and its subareas according to a study conducted by Siddarth P et al. [6]. It is unknown what mechanism underpins the relationship between sedentary behavior and MTL thickness. Sedentary behavior appears to directly impact neurobiological processes. Sedentary behavior is shown to harm the brain by minimizing neurogenesis, synaptic plasticity, neurotrophin generation, angiogenesis, and inflammation, which are pathological mechanisms that impact hippocampal integrity.

In contrast to recent findings, our data finds evidence of a null connection of PA with the right hippocampus, parahippocampal, posterior cingulate cortex, amygdala, TIV, GMV, and WMV. Two factors may explain these disparities in results: first, cross-cultural differences; second, the study’s cross-sectional design and the technique used to evaluate PA.

In the present study, we’ve obtained the PA characteristics of participants using the IPAQ, which gives subjective measurement. Due to the use of self-perceived PA in observational studies, which is often inexplicable, our current understanding of the relationship between PA and brain volume remains speculative, necessitating more research using objectively measured PA.

The causal effect due to PA and sedentary behavior on brain volume cannot be inferred because of the cross-sectional nature of the study. Many previous studies have included follow-up of participants after a few years from baseline data collection and change in brain volume was assessed over a period of time [26,31,32]. Following this strategy might give a clearer understanding of the effect of lifestyle factors on brain volume.

The current study’s findings were not clinically correlated. Some lifestyle choices like dietary habits could have an effect on these findings; therefore, future research should investigate and account for these. Studies like this one, which show links between PA and neuroimaging measures, are an essential step toward understanding how physical exercise and sedentary behaviors affect the brain.

### Limitations and Recommendations

Since we could not achieve the desired sample size, we could not find a significant association between brain ROIs (except for the left hippocampus) and brain volumes with PA and SB. Further studies with a larger sample size might demonstrate a significant association between PA, SB, and brain volumes.

Since PA measures in the study were self-reported, there are chances of recall and response bias. Future studies can be done using objectively measured PA instead of a self-reported questionnaire.

## 5. Conclusions

Total physical activity is strongly associated with the left hippocampal volumes. Except for the hippocampus, the current study found no significant association between physical activity and other brain structures. Before we can confidently say that engaging in planned PA is a viable approach for dissipating age-related brain volume losses, particularly cognition-related ROIs, and promoting effective brain aging at the population level, the association between physical activity, sedentary behavior, and brain volume needs to be investigated further.

## Figures and Tables

**Figure 1 ijerph-19-17079-f001:**
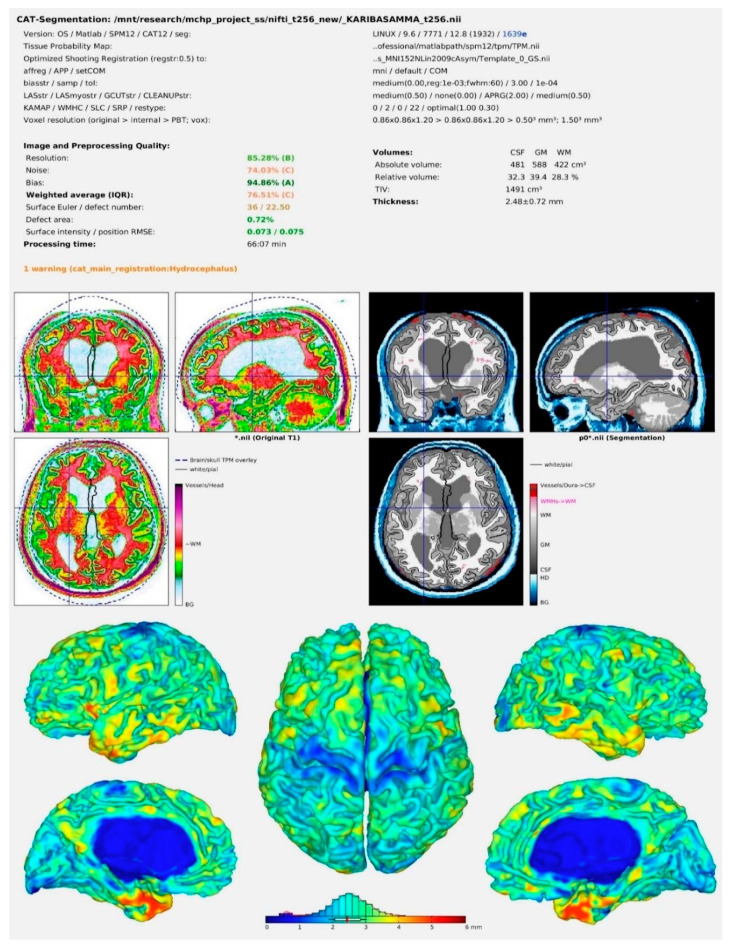
CAT12 result output image after pre-processing of the MRI scans.

**Figure 2 ijerph-19-17079-f002:**
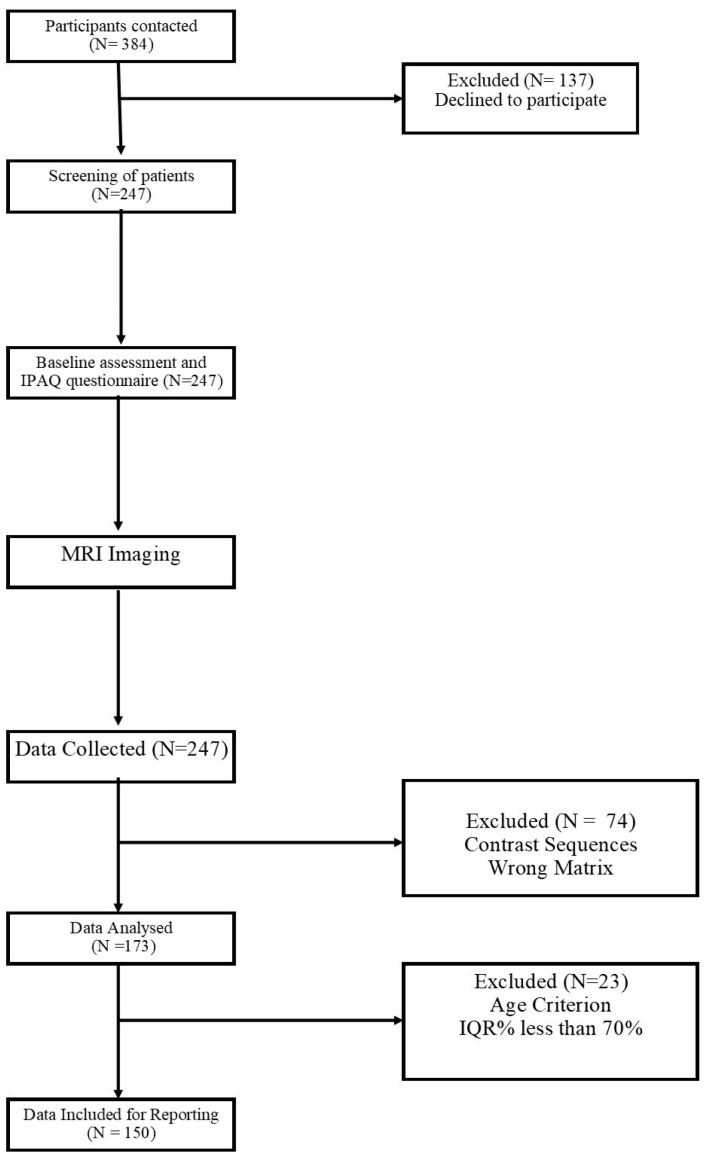
STROBE flow diagram.

**Table 1 ijerph-19-17079-t001:** Demographic, Cardiovascular risk factors, and Lifestyle factors of the patients.

Demographics	Total (*n* = 150)
Age group	Below 20 years *n* = 9
	20–30 years *n* = 32
	30–40 years *n* = 31
	40–50 years *n* = 39
	50–60 years *n* = 39
Mean age based on physical activity levels	Low physical activity = 39.59
	Moderate physical activity = 40.01
	Vigorous physical activity = 31.6
Gender (*n*)	Male *n* = 86; Female *n* = 64
Height mean (SD)	162.1 (6.73)
Weight mean (SD)	62.53 (11.78)
Education *n* (%)	Schooling = 83 (55.3%)
	Graduate = 61 (41.3%)
	Post-graduate = 5 (3.3%)
Employment status *n* (%)	Unemployed *n* = 77 (52%)
	Employed *n* = 73 (48%)
Job experience *n* (%)	More than 5 years = 63 (42%)
	Less than 5 years = 9 (6%)
	Not applicable = 78 (52%)
Cardiovascular risk factors	
BMI mean (SD)	23.7 (4.15)
Waist circumference mean (SD)	33.2 (2.9)
Mean arterial pressure mean (SD)	93.3 (6.59)
SBP mean (SD)	122.07 (10.63)
DBP mean (SD)	78.91 (5.66)
T2DM *n* (%)	6 (4%)
Hypertension *n* (%)	12 (8%)
Medications *n* (%)	(9.3%)
Lifestyle factors	
Alcohol consumption *n* (%)	9 (6%)
Smoking *n* (%)	4 (2.67%)

**Table 2 ijerph-19-17079-t002:** Association between brain volume, physical activity, and sedentary behavior.

Brain Volume	Vigorous (β, *p* < 0.05)		Moderate (β, *p* < 0.05)		Light (β, *p* < 0.05)		TPA (β, *p* < 0.05)		Sitting (β, *p* < 0.05)	
**Left hippocampus**	**−0.328**	**0.049 ***	**0.04**	**0.813**	−0.007	0.97	0.710	**0.021 ***	−0.053	0.586
**Right hippocampus**	−0.207	**0.236**	−0.03	0.236	−0.018	0.929	0.564	**0.079**	−0.002	0.988
**Left PCC**	−0.235	0.173	0.173	0.323	0.206	0.314	0.311	0.322	−0.01	0.922
**Right PCC**	−0.227	0.189	0.116	0.508	0.08	0.696	0.368	0.244	−0.006	0.95
**Left AMYG**	−0.265	0.112	0.143	0.398	0.043	0.828	0.505	0.099	−0.101	0.303
**Right AMYG**	−0.006	0.971	0.075	0.666	−0.004	0.986	0.354	0.258	−0.007	0.944
**Left PHG**	−0.227	0.164	0.143	0.387	−0.039	0.839	0.564	0.06	−0.066	0.493
**Right PHG**	−0.126	0.447	0.129	0.446	0.017	0.931	0.424	0.164	0.022	0.82
**Total volume**	−0.04	0.81	0.088	0.603	−0.071	0.721	0.314	0.303	−0.05	0.609
**Gray matter volume**	−0.175	0.286	0.093	0.575	0.002	0.99	0.475	0.388	−0.04	0.677
**White matter volume**	−0.067	0.695	0.061	0.695	0.01	0.961	0.388	0.213	−0.091	0.369

Note: Values are β, *p*-values. β—Standard regression coefficient; * indicates a significant association. PCC—Posterior cingulate cortex, AMYG—Amygdala, PHG—Parahippocampal gyrus. TPA—Total Physical Activity.

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
