# Peer review of "Association between Physical Activity Levels and Brain Volumes in Adults Visiting Radio-Imaging Center of Tertiary Care Hospital"

_ijerph, 2022, doi:10.3390/ijerph192417079_

Round 1
Reviewer 1 Report
The aim of article titled: Association between physical activity levels and brain volumes in adults visiting radio-imaging center of tertiary care hospital was to explore the relationship between physical activity and sedentary behavior on brain volume.
Introduction section include a lot of definitions/expressions rather than an explanation of the reason to conduct the study. The author’s main focus was put on cognitive impairments and Alzheimer’s disease while the main aim was to explore the relationship between physical activity and sedentary behavior on brain volume. What is the link between cognitive function and brain volume? Not much in that matter was discussed.
Also the age of group was very divergent. What was the mean age and SD in the group. Did the authors check what was the mean age in the physical activity categories?
There are fragments of the manuscript that lack relevant references, for example Line 191: “Various observational epidemiological studies have investigated the potential relationship between physical activity and cognitive decline…” from where do the authors state that? No citation included.
There is a lot of studies included concerning cognitive function. It raises up a question: Did the authors check cognitive function of the participants?
There is a lot of abbreviations in the text that were not explained: Line 21-22 in the abstract section, Line 57, 127 in the manuscript. Please explain the abbreviations.
Author Response
Dear reviewer,
We are grateful for the valuable reviews provided by the peer review team. We have made the necessary changes and highlighted the same. We have provided a detailed changes in the table below for your reference
Looking forward to the favourable review
|
Details of the suggested changes |
|||||
|
Association between Physical Activity Levels and Brain Volumes in Adults Visiting Radio-Imaging Center of Tertiary Care Hospital
|
|
||||
|
Reviewer |
SL.NO |
Suggestions received |
Clarifications |
Line number |
|
|
1 |
1 |
Introduction section include a lot of definitions/expressions rather than an explanation of the reason to conduct the study. The author’s main focus was put on cognitive impairments and Alzheimer’s disease while the main aim was to explore the relationship between physical activity and sedentary behavior on brain volume. What is the link between cognitive function and brain volume? Not much in that matter was discussed. |
The suggested changes has been incorporated |
33 - 58
|
|
|
2 |
Also the age of group was very divergent. What was the mean age and SD in the group. Did the authors check what was the mean age in the physical activity categories? |
The mean age calculations based on the physical activity has been added |
Table 1 |
||
|
3 |
There are fragments of the manuscript that lack relevant references, for example Line 191: “Various observational epidemiological studies have investigated the potential relationship between physical activity and cognitive decline…” from where do the authors state that? No citation included. |
The references has been added and highlighted |
Line 181 |
||
|
4 |
There is a lot of studies included concerning cognitive function. It raises up a question: Did the authors check cognitive function of the participants? |
There was no test were made to assess the cognitive functions since our study aim was to assess the brain regions volume in SB and PA lifestyle. So assessing cognitive functions are out of the scope of this study |
|||
|
|
5 |
There is a lot of abbreviations in the text that were not explained: Line 21-22 in the abstract section, Line 57, 127 in the manuscript. Please explain the abbreviations. |
The changes have been highlighted |
21-23 |
|
Reviewer 2 Report
This is an interesting study that explores the effects of different exercise combinations on the cognitive and physical function. This manuscript is well written and easy to follow. Only a minor revision is suggested to this manuscript.
1. Abstract: the abbreviations such as TIV, GMV, and WMV also should be explained or readers can not unstand; the word "Anecdotal evidence" is suggested to substitute in scientific papers. Besides, the number following the decimal point should be consistent. Please check the whole paper.
2. Introduction: there is a need to add a piece of information on the association between physical activity or fitness and cognitive function (see doi: 10.1186/s12877-022-03564-9) to the Introduction. Further, the hypothesis should be provided after the aim of study.
3. Methods: brain volume may be associated with aging. Why the authors recruited participants with a wide range of ages (18-60 years)? Please provide information about who conducted the test and analyse.
4. Results: there are many formatting errors in Tables such as missing space and the number following the decimal point is inconsistent, so please check the Tables and the whole paper carefully.
5. Discussion: please add a paragraph to explain the possible mechanisms between PA and brain volume.
6. Conclusion: please delete P values from the section.
Author Response
Dear reviewer,
We are grateful for the valuable reviews provided by the peer review team. We have made the necessary changes and highlighted the same. We have provided a detailed changes in the table below for your reference
Looking forward to the favourable review
|
2 |
1 |
Abstract: the abbreviations such as TIV, GMV, and WMV also should be explained or readers can not unstand; the word "Anecdotal evidence" is suggested to substitute in scientific papers. Besides, the number following the decimal point should be consistent. Please check the whole paper |
The suggested changes has been incorporated
|
21-23
17 |
|
|
2 |
Introduction: there is a need to add a piece of information on the association between physical activity or fitness and cognitive function (see doi: 10.1186/s12877-022-03564-9) to the Introduction. Further, the hypothesis should be provided after the aim of study. |
The suggested changes has been incorporated
|
|
|
|
3 |
Methods: brain volume may be associated with aging. Why the authors recruited participants with a wide range of ages (18-60 years)? Please provide information about who conducted the test and analyse |
We used a wide age group range to do a sub group analysis. The subgroup analysis will be performed in the future studies and these confounding factors will be included |
|
|
|
4 |
Results: there are many formatting errors in Tables such as missing space and the number following the decimal point is inconsistent, so please check the Tables and the whole paper carefully. |
The suggested changes has been incorporated
|
Table 2 |
|
|
5 |
Discussion: please add a paragraph to explain the possible mechanisms between PA and brain volume |
The possible mechanism is explained |
183-187 |
|
|
6 |
Conclusion: please delete P values from the section |
The suggested changes has been incorporated
|
|
Reviewer 3 Report
This is a study on the relationship between daily physical activity obtained from questionnaire and brain volume assessed by 1.5T MRI. The thema seems within the scope of this journal and the overall study design and the manuscript seems satisfactory, although several intrinsic limitations should be carefully evaluated.
1. About the cohort of the study.
The number of participants is relatively small with only 13 are categolized in the high physical activity group. The range of age is rather wide from below 20 to 60 years with both sex mixed (as well as both dominant hand, job status etc.), suggesting many potential confounding factors exist in this cohort.
2. left vs right hippocampus.
The data showed left but not right hippocampus showed significant association with physical activity, but data on the right side also showed almost significant level (P=0.079; TPA). left side significancy would be relative and this reviewer advise to mention on this point.
3. abbreviation.
Some are not shown in the manuscript (for example, TPA, PCC)
Author Response
Dear Reviewer,
We are grateful for the valuable reviews provided by the peer review team. We have made the necessary changes and highlighted the same. We have provided a detailed changes in the table below for your reference
Looking forward to the favourable review
|
3 |
1 |
The number of participants is relatively small with only 13 are categorized in the high physical activity group. The range of age is rather wide from below 20 to 60 years with both sex mixed (as well as both dominant hand, job status etc.), suggesting many potential confounding factors exist in this cohort |
We used a wide age group range to do a sub group analysis. The subgroup analysis will be performed in the future studies and these confounding factors will be included |
|
|
|
2 |
left vs right hippocampus. The data showed left but not right hippocampus showed significant association with physical activity, but data on the right side also showed almost significant level (P=0.079; TPA). left side significancy would be relative and this reviewer advise to mention on this point.
|
To state any value to be statistically significant, the p value has to be less than 0.05 but the right hippocampus value is slightly more than that so we did not emphasis on that but the value has been highlighted |
Table 2 |
|
|
3 |
abbreviation. Some are not shown in the manuscript (for example, TPA, PCC)
|
The suggested changes has been incorporated
|
Table 2
|
Round 2
Reviewer 1 Report
Thank you for the corrections.